# Genetic Loci Mining and Candidate Gene Analysis for Determining Fatty Acid Composition in Rice

**DOI:** 10.3390/genes15111372

**Published:** 2024-10-25

**Authors:** Yiyun Ge, Yiting Wei, Xuan Li, Zhenan Zhu, Jinjin Lian, Huimin Yang, Tiantian Lu, Sanfeng Li, Jiahui Huang, Yuhan Ye, Yuexing Wang, Yuchun Rao

**Affiliations:** 1College of Life Sciences, Zhejiang Normal University, Jinhua 321004, China; 15105816895@163.com (Y.G.); 17865331245@163.com (Y.W.); lx20002020@zjnu.edu.cn (X.L.); 13705870600@163.com (Z.Z.); lianjin0829@163.com (J.L.); 19519802611@163.com (H.Y.); 18969367029@163.com (T.L.); jhhuang08@163.com (J.H.); yeyuhan70@gmail.com (Y.Y.); 2State Key Laboratory of Rice Biology and Breeding, China National Rice Research Institute, Hangzhou 311400, China; lisanfeng@cass.cn

**Keywords:** rice, fatty acid, QTL mapping, germplasm resources

## Abstract

Fatty acid composition and its proportions are critical to the nutritional value and storage quality of rice (*Oryza sativa* L.) as the third major nutrient component in this staple food. This study involved crossing an indica rice variety, Huazhan (HZ), as the male parent, with a japonica variety, Nekken2, as the female parent, to produce the F1 generation. Subsequently, a population of 120 recombinant inbred lines (RILs) was developed through multiple generations of self-breeding. By utilizing a high-density molecular genetic linkage map and phenotypic data of four fatty acid components, we identified a total of 14 quantitative trait loci (QTLs) related to fatty acid composition across chromosomes 1, 3, 4, 6, 8, and 9. These included two QTLs for C_14_ content, three for C_16:0_ content, six for C_18:1_ content, and three for C_18:2_ content. Notably, the QTL *qCOPT4.2* exhibited a high LOD score of 5.22. Within QTL intervals, genes such as *OsACX3* and *SLG* affecting grain length were identified. The expression of candidate genes within these intervals was assessed and further analyzed by using quantitative real-time PCR. Genes such as *LOC_Os01g15000*, *LOC_Os04g47120*, *LOC_Os04g49194*, *LOC_Os06g22080*, *LOC_Os06g23870*, *LOC_Os06g24704*, *LOC_Os06g30780*, *LOC_Os08g44840*, and *LOC_Os09g36860* were found to regulate fatty acid synthesis or metabolic pathways, potentially enhancing fatty acid content in rice. These QTLs are indispensable for breeding rice varieties with improved fatty acid profiles, offering new genetic resources for enhancing the nutritional and storage qualities of rice.

## 1. Introduction

“Food is the primary necessity of the people”. This adage underscores the essential role of food in human life. As the standard of living improves in China, consumer demands have evolved from merely “eating enough” to “eating well”. In alignment with the “Healthy China 2030” planning outline, the “National Nutrition Plan” was formulated to elevate the nutritional health of the population by intensifying foundational research and enhancing the nutrition of staple foods [1].

Rice serves as the staple food for nearly 65% of the Chinese population, making it the country’s most crucial crop [2]. Therefore, improving rice quality represents an effective strategy for enhancing the nutritional structure of the national diet, safeguarding public health, and ensuring efficient nutrient intake from staple foods. It also significantly contributes to increasing the value-added properties of grain crops and satisfying the demands of the food consumption market [3].

The fat content in rice grains is a key indicator of their nutritional quality. As an important component of rice’s nutritional profile, fatty acids determine the fat content and, by extension, the overall quality of the rice grains [4]. Modulating fatty acid content is thus an important method for improving the edibility, cooking quality, and storage characteristics of rice [5]. In plants, fatty acids are synthesized in plastids from acetyl-CoA and stored as oil bodies, supplying energy for plant growth and development through fatty acid metabolic pathways [6,7]. The fatty acid content in rice positively correlates with its nutritional value, appearance, storage quality, and other parameters, such as gel consistency, gelatinization temperature, and water absorption rate [8]. Consequently, enhancing the fatty acid content in rice can improve the texture and nutritional benefits of rice meals.

The primary fatty acids in rice include myristic acid (C_14_), palmitic acid (C_16:0_), oleic acid (C_18:1_), and linoleic acid (C_18:2_), which provide energy and essential fatty acids that cannot be synthesized by the body. These are oxidized through fatty acid metabolic pathways in the body [9,10]. Research indicates that consuming unsaturated fatty acids, such as linoleic acid and palmitic acid, aids in improving blood circulation and preventing cardiovascular diseases [11].

Despite the high quality and nutritional value of the unsaturated fatty acids in rice, they comprise only a small fraction (0.3–4%) of the three main nutritional components of rice and are difficult to measure. Consequently, studies on rice fatty acid-related quantitative trait loci (QTLs) and related genes have been relatively sparse [12]. In the 1990s, Kang and colleagues identified a QTL controlling rice fatty acid content on chromosome 7 within the KCD405-RZ395 interval by using an MGRIL recombinant inbred line population [13]. Subsequent studies have detected several QTLs with significant additive effects on crude fat content in brown rice, including *qLc-5*, which showed the largest effect and originated from the maternal parent in the study population [14]. Further efforts using Chromosome Segment Substitution Lines (CSSLs) have identified stable QTLs across different growth environments, demonstrating the potential for precise QTL mapping and gene cloning [15]. Tong and others performed preliminary association analysis on lysophospholipid (LPL) content in 13 rice germplasm varieties, identifying key QTLs linked to LPL biosynthesis and developing Indel molecular markers for functional markers derived from candidate genes [16].

Although significant progress has been made in mapping QTLs that control rice fatty acid content, the exploration and analysis of the genes regulating this trait are not yet extensive, and the molecular genetic mechanisms remain unclear. Current research predominantly focuses on mutant locus cloning and homology identification, which tend to be superficial. Presently, by screening for major-effect QTLs and principal genes related to rice fatty acid content and employing molecular marker-assisted breeding techniques, it is feasible to cultivate rice varieties with enhanced fatty acid content, thereby boosting their nutritional value and storage quality. This study leverages a recombinant inbred line (RIL) genetic population developed from Huazhan (HZ) and Nekken2. By sampling grains from both parental lines and the population at maturity and performing fatty acid content and composition analysis, we aim to elucidate QTLs related to rice fatty acid content, discover more candidate genes, and establish a foundation for breeding high-fatty acid rice varieties while providing new germplasm resources to enhance the nutritional value and storage quality of rice.

## 2. Materials and Methods

### 2.1. Plant Materials

This study utilized the indica rice variety Huazhan (*Oryza sativa* L. subsp. *indica* cv. “HZ”) as the paternal parent and the japonica rice variety Nekken2 (*Oryza sativa* L. subsp. *japonica* cv. “Nekken2”) as the maternal parent to obtain the F1 generation through hybridization. Single-grain propagation based on bagging and self-crossing over multiple generations resulted in 120 recombinant inbred lines (RILs) with stable phenotypic and genotypic inheritance, thereby establishing the RIL genetic population [17]. It is worth noting that the materials “Huazhan”, “Nekken2”, and 120 RILs mentioned above were provided by the China Rice Research Institute. “Huazhan” is a conventional indica rice variety bred by the China Rice Research Institute and the Rice Research Institute of Guangdong Academy of Agricultural Sciences. It has good growth characteristics and adaptability. As a japonica conventional rice bred in Japan, “Nekken2” has broad-spectrum affinity and strong affinity. Therefore, we chose these two varieties as research materials.

### 2.2. Experimental Methods

#### 2.2.1. Seed Germination and Cultivation

A random selection of 100 uniform, well-formed, and healthy seeds from both parents and RILs was made. Seeds were initially soaked and cleaned in a 70% ethanol solution for 2 min, which was repeated three times; they were then disinfected in a 10% sodium hypochlorite solution for 10 min and subsequently rinsed with deionized water to remove any residual disinfectant [18]. After disinfection, the seeds were placed in kraft paper bags and submerged in water for 48 h (changing the water every 24 h), followed by germination in moist cheesecloth at 37 °C in a constant temperature incubator for 48 h.

#### 2.2.2. Rice Planting and Management

Seedlings with vigorous growth and consistent appearance were selected for transplanting onto a nursery bed for seeding and cultivation. Once the seedlings reached the 3–4-leaf stage, uniformly grown seedlings were transplanted and arranged in a 4-row-by-6-column pattern. Regular field management practices including timely weeding and pest control were maintained.

#### 2.2.3. Preparation of Rice Flour

After maturation, grains from both parents and each RIL were collected, air-dried, and stored at 4 °C in a seed storage cabinet. The rice grains were processed into rice flour in four steps: hulling with a yield tester (JDMZ 100 type, Suzhou, China), milling with a rice miller (JNMJ3 type, Beijing, China), grinding with a cyclone mill (FS-II type, Hangzhou, China), and sifting with a standard test sieve (GB/T 6003.1-2012 type, Guangdong, China). The prepared rice flour samples were then packaged by lineage into 10 × 6 cm kraft paper bags for subsequent use.

#### 2.2.4. Fatty Acid Content Determination and Analysis

Preparation of extraction and internal standard solutions: A mixture of 2 L of methanol (CH_3_OH) (MACKLIN) and 40 mL of 2% sulfuric acid (H_2_SO_4_) (Sinopharm, Beijing, China) was prepared, to which 0.2 g of 0.01% butylated hydroxytoluene (BHT) (Sinopharm) was added to create the rice fatty acid extraction solution; a total of 400 mg of margaric acid methyl ester (C_17:0_) (BePure, Beijing, China) was dissolved in 200 mL toluene (C_7_H_8_) (Sinopharm) to prepare the fatty acid internal standard solution (C_17:0_). Fatty acid content was determined by weighing 0.2 g of rice flour from each parent and RIL into a 15 mL test tube. In a fume hood, 4.5 mL of rice fatty acid extraction solution and 400 µL of the fatty acid internal standard solution were added, and the tubes were flushed with nitrogen, tightly capped, and heated in an 85 °C water bath for 2 h, with agitation every 30 min. After heating, the tubes were cooled to room temperature, and 2 mL of hexane (Vokai, Beijing, China) was added and mixed thoroughly. The mixture was then centrifuged at 4 °C and 1000 rpm for 5 min, and 600 µL of the supernatant was transferred to a sample vial, which was left at room temperature. The sample was then injected into a chromatography column (Agilent HP-5 capillary column: 30 m long, 0.32 mm in inner diameter, and 0.25 µm in film thickness, Santa Clara, CA, USA), with the column oven’s initial temperature set to 150 °C, and the column temperature and flow rate accurately adjusted to detect the content of various fatty acids.

#### 2.2.5. Construction of Genetic Maps

DNA was sampled and extracted from both parents and the RIL population, purified, and whole-genome-sequenced. Data were organized and analyzed to obtain 4858 molecular markers evenly distributed across all 12 rice chromosomes, used to construct a corresponding high-density genetic linkage map.

#### 2.2.6. QTL Mapping and Analysis

By utilizing previously established high-density molecular marker genetic linkage maps, fatty acid-related component data were processed by using Map-maker/QTL1.1B software for QTL mapping and composite interval mapping method. The LOD threshold was set to 2.5; a QTL was considered present if an LOD score greater than 2.5 was detected between two markers at the location with the highest LOD value. QTL naming adhered to the principles proposed by Mccouch et al. [19].

#### 2.2.7. Analysis of Candidate Gene Expression for Fatty Acid Content

Following the protocol of a total RNA extraction kit, RNA from mature fresh grains of both parents was extracted and reverse-transcribed into cDNA by using an RNA reverse transcription kit. Based on the QTL interval analysis results, candidate genes related to rice fatty acid content were selected for real-time quantitative PCR (qRT-PCR) analysis, using rice *OsActin* as the internal reference gene. The expression differences of candidate genes between the parents were detected and analyzed, and quantitative results were processed by using the 2^−ΔΔCT^ method [20]. Excel and GraphPad Prism were used for data analysis and graphing, with experimental data differences being compared by using the *t*-test; in the *t*-test results, a *p*-value less than 0.05 indicated that there was a significant difference in the expression level of the candidate gene between the parents, which is represented by “*” in the result figure. A *p*-value is less than 0.01 indicated a highly significant difference in the expression level of the candidate gene between the parents, which is indicated by “**” in the result figure.

qRT-PCR reaction system: 1 µL of cDNA template, 0.4 µL each of forward and reverse primers (10 µmol/L), 5 of µL SYBR qPCR Mix (Toyobo, Tokyo, Japan), and 3.2 µL of ddH_2_O briefly centrifuged to mix.

qRT-PCR amplification program: 95 °C for 5 min, 95 °C for 10 s, 57 °C for 20 s, and 72 °C for 20 s, for a total of 40 cycles. Related primer sequences are shown in Table 1.

## 3. Results

### 3.1. Fatty Acid Content Characteristics of Parents and Recombinant Inbred Line Population

The analysis of fatty acid content in the grains of the parent varieties (HZ and Nekken2) revealed significant differences in the levels of myristic acid (C_14_), palmitic acid (C_16:0_), oleic acid (C_18:1_), and linoleic acid (C_18:2_) (see Figure 1). Myristic acid content was 0.3397 µg/g in HZ and 0.8225 µg/g in Nekken2; palmitic acid was 9.3513 µg/g in HZ and 24.9255 µg/g in Nekken2; oleic acid was 13.2595 µg/g in HZ and 14.1534 µg/g in Nekken2; linoleic acid was 8.3246 µg/g in HZ and 32.3430 µg/g in Nekken2. These results indicate that Nekken2 has significantly higher fatty acid content than HZ, suggesting that Nekken2 grains possess higher nutritional value and superior storage quality.

In the RIL population, the concentrations of myristic acid (C_14_), palmitic acid (C_16:0_), oleic acid (C_18:1_), and linoleic acid (C_18:2_) showed a continuous normal distribution and were widely varied, including many instances of transgressive segregation. This indicates that fatty acid content in this population is influenced by multiple quantitative trait loci, meeting the criteria for QTL interval mapping.

### 3.2. QTL Mapping Analysis for Rice Fatty Acid Content

By using the 14,858 SNP high-density molecular marker genetic map previously developed from the RIL population, 14 QTLs related to fatty acid components were detected across chromosomes 1, 3, 4, 6, 8, and 9 (Figure 2; Table 2), including 2 QTLs related to myristic acid (C_14:0_), 3 to palmitic acid (C_16:0_), 6 to oleic acid (C_18:1_), and 3 to linoleic acid (C_18:2_). Notably, QTL clusters on chromosomes 4 and 6 concurrently influenced the four traits associated with fatty acid content, including C_14_, C_16:0_, C_18:1_, and C_18:2_. On chromosome 4, the QTLs *qCOFF4*, *qCOFS4*, *qCOPO4*, and *qCOPT4.2* significantly overlapped, with LOD scores of 4.05, 4.56, 5.04, and 5.22, respectively; these represented the highest LOD scores in this mapping effort. The QTLs *qCOFF6*, *qCOFS6.2*, *qCOPO6*, and *qCOPT6* on chromosome 6 also showed significant overlap. These results indicate that the identified QTL clusters are major-effect clusters governing rice fatty acid content, and the genes involved may exhibit pleiotropy or linkage. This finding highlights that the synthesis and metabolism of rice fatty acids are regulated by complex quantitative trait genes. Among the mapped QTLs, *qCOPT4.2* had the highest LOD score, located between 123 and 132 cM on chromosome 4, with a peak LOD of 5.22, suggesting the presence of a major gene regulating rice grain fatty acid content within this interval.

### 3.3. Expression Analysis of Candidate Genes Related to Rice Fatty Acid Content

Based on the locations of fatty acid-related QTL mapping intervals on the chromosomes, and in conjunction with the rice genome annotation portal (http://rice.plantbiology.msu.edu/ accessed on 24 January 2024), functional genes within the QTL intervals were identified and assessed. A preliminary summary of these genes’ functions and cloning status is provided in Table 3.

Candidate genes associated with fatty acid content include those encoding enzymes and proteins such as lipase, acyl-CoA thioesterase 2, flavanone 3-hydroxylase, diacylglycerol O-transferase, acyl-CoA dehydrogenase domain protein, the OsACX3 energy gene, the acyl desaturase chloroplast precursor, the SLG grain length gene, and the acyl carrier protein gene. The real-time quantitative PCR analysis of nine candidate genes related to fatty acid content in mature grains of both parents revealed significant upregulation in Nekken2 compared with HZ; notably, LOC_Os06g23870 and LOC_Os08g44840 showed particularly pronounced increases (Figure 3). These findings suggest that these candidate genes may influence the composition and levels of fatty acids in rice grains by regulating pathways involved in fatty acid synthesis and metabolism, thereby affecting the nutritional quality of rice.

## 4. Discussion

Fatty acids in rice grains serve as essential nutrients, providing energy and essential fatty acids required for the growth of both animals and plants, which these organisms cannot synthesize independently. Beyond their role as functional molecules and critical components of biological membranes, fatty acids represent a significant source of energy in the diet [21]. The classification, biosynthesis, and metabolic pathways of fatty acids are diverse and intricate. In rice, fatty acid synthesis contributes to responses to both biotic [22] and abiotic stresses [23] and involves complex molecular regulatory networks. The trait of fatty acid content in rice is quantitatively controlled by several genes, the expression of which can show large differences depending on population selection, growth environment and growth cycle, thus affecting the QTL mapping results [16]. Therefore, it is necessary to select a stable and reliable population of RILs for QTL determination. Genetic effects have a significant impact; integrating QTL mapping with genes that have major effects on the regulation of fatty acid content in rice grains and selectively applying these genes in rice breeding can accelerate the development of high fatty acid rice varieties. This approach promotes improvements in the nutritional value and storage quality of rice.

Real-time quantitative PCR confirms that fatty acid content in rice is a quantitatively controlled trait influenced by multiple genes, with environmental factors, intrinsic factors, and human selection impacting the expression of relevant candidate genes, thus affecting the fatty acid content in rice grains. The selection of the population, the natural growth environment, and the methods used for fatty acid extraction and analysis can all impact QTL mapping results to varying degrees. This study focuses on four fatty acid components (C_14_, C_16:0_, C_18:1_, and C_18:2_) as observational indicators. Based on the results of QTL mapping and real-time quantitative PCR analysis of candidate genes, this study identifies genes that may regulate fatty acid content, providing reference data for breeding high-fatty acid-content rice varieties and offering new germplasm resources to enhance the fatty acid content, nutritional value, and storage quality of rice grains.

This research study identified a total of 14 QTLs associated with four distinct fatty acid content components, specifically C14, C16:0, C18:1, and C18:2. Notably, QTL clusters located on chromosomes 4 and 6 were found to regulate all four fatty acid components simultaneously. Fengying Xue [24] conducted a genome-wide association study (GWAS) using resource materials collected from 2018 to 2019, mapping QTLs pertinent to fatty acid content. In her findings, the QTL qLC2018N4.1 was located in close proximity to the qCOPT4.1 interval identified in the present study, which pertains to the C18:2 component. Additionally, qLC2018N4.2 was found to be very near the QTL cluster on chromosome 4 that influences all four fatty acid components.

The QTL cluster on chromosome 6, which encompasses qCOFF6, qCOFS6.2, qCOPO6, and qCOPT6 (spanning from 37.83 cM to 90.65 cM), was observed to partially overlap with the QTL QFC-6, as identified by Shen et al. [25] in a recombinant inbred line (RIL) population derived from the cross of Sasanishiki and Habataki. This QTL accounted for 16.33% of the phenotypic variation in fatty acid content and exhibited a logarithm of odds score of 6.14, indicating its significant effect on the trait. These findings collectively reinforce the existence of a major-effect QTL site at these loci that governs fatty acid content, highlighting its potential importance for future breeding strategies aimed at optimizing fatty acid profiles in the studied population.

Additionally, several new loci related to fatty acid content were identified on chromosomes 1, 3, 6, 8, and 9, each with LOD scores around 3, suggesting the probable presence of major genes regulating fatty acid content traits within these intervals. In the *qCOPO1* interval, *LOC_Os01g15000*, encoding a lipase, was identified; research by Worawat et al. [26] confirmed that this gene is a class III lipase gene involved in energy metabolism in response to salt stress. In the *qCOPO8* interval, a gene encoding a BAHD acyltransferase, *LOC_Os08g44840*/*SLG*, which regulates rice grain size and leaf angle by mediating the balance of brassinosteroids within the rice plant, was identified [27]. In the new QTL interval *qCOPO9*, the gene encoding acyl carrier protein, *LOC_Os09g36860*, was also identified; research by Hu et al. [28] based on the transcriptomic analysis of cytoplasmic male sterility lines suggests that this gene may be involved in regulating the lipid composition of pollen during pollen development. The functional genes identified in these new intervals are related to traits such as lipases, grain size, and pollen development, suggesting that more major-effect genes exist within these intervals that participate in regulating fatty acid content, influencing rice growth and development.

Moreover, in the *qCOPT4.1* interval adjacent to *qLC2018N4.1* identified by Xue [24], *LOC_Os04g47120* has been confirmed in proteomics studies as an acyl-CoA thioesterase 2 (peroxisomal enzyme) gene, providing energy for jasmonic acid biosynthesis, auxiliary β-oxidation, and detoxification [29]. In the QTL cluster on chromosome 4 that controls all four traits, *LOC_Os04g49194* participates in the biosynthesis of flavonoid secondary metabolites, playing a crucial role in energy provision during plant–microbe symbiosis [30]. The QTL cluster on chromosome 6 that overlaps with *QFC-6* identified by Shen et al. [25] included the gene encoding acetyl-CoA oxidase *LOC_Os06g24704*/*OsACX3*, an energy gene. Research by Kim et al. [31] in 2007 confirmed that *OsACX3* shows differential β-oxidation activity during plant development, with the expression of this gene being involved in providing energy for rice plants during the late stages of seedling development. Fatty acid biosynthesis primarily occurs in plastids, where diacyl-ACP thioesterase can catalyze the inhibition of fatty acid growth, and *LOC_Os06g22080* regulates its own gene expression to participate in the biosynthesis or degradation of fatty acids [32]. The genome-wide analysis of a novel fatty acid desaturase gene provides a reference for the molecular regulatory network of unsaturated fatty acid metabolism and signal transduction pathways in rice [22]. *LOC_Os06g30780* is an acyl desaturase (chloroplast precursor) gene, identified by Biswas et al. [33] in the *qCTSL-6-1* interval during QTL mapping, and is hypothesized to be responsible for cold tolerance in rice.

Therefore, uncovering more QTLs related to fatty acid content and desaturase function gene analysis and cloning will aid scientists in their breeding research on fatty acids in rice grains. Furthermore, the cloning of candidate genes related to fatty acid content is crucial to understanding the molecular regulatory networks involved in the biosynthesis or degradation pathways of fatty acids in rice grains. Applying these genes to transgenic food crops can produce grains with high nutritional value and health benefits and high in fatty acid content, which holds practical significance for human health.

## Figures and Tables

**Figure 1 genes-15-01372-f001:**
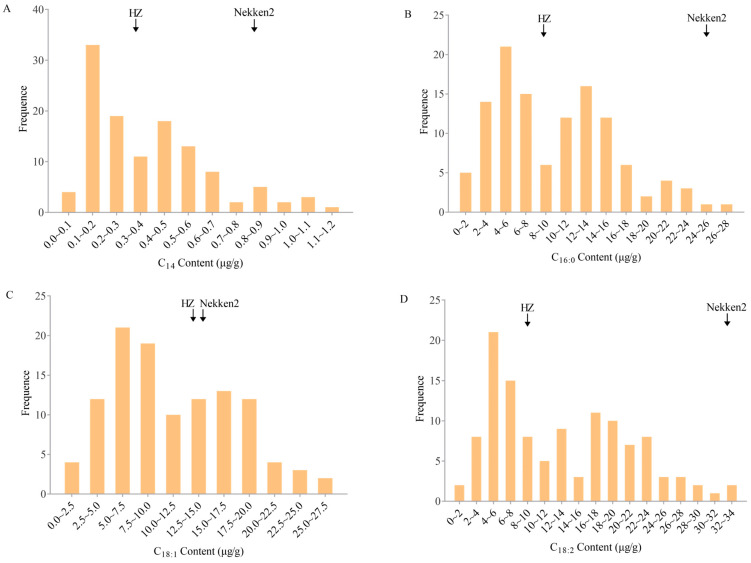
The contents of fatty acids in rice RIL population. (**A**) myristate (C14), (**B**) palmitic acid (C16:0), (**C**) oleic acid (C18:1), and (**D**) linoleic acid (C18:2) were normally distributed.

**Figure 2 genes-15-01372-f002:**
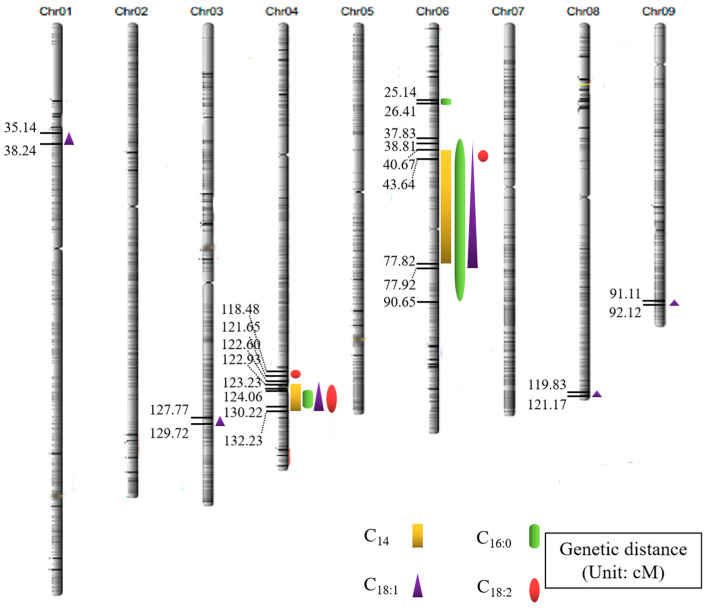
QTL mapping of fatty acid content in rice.

**Figure 3 genes-15-01372-f003:**
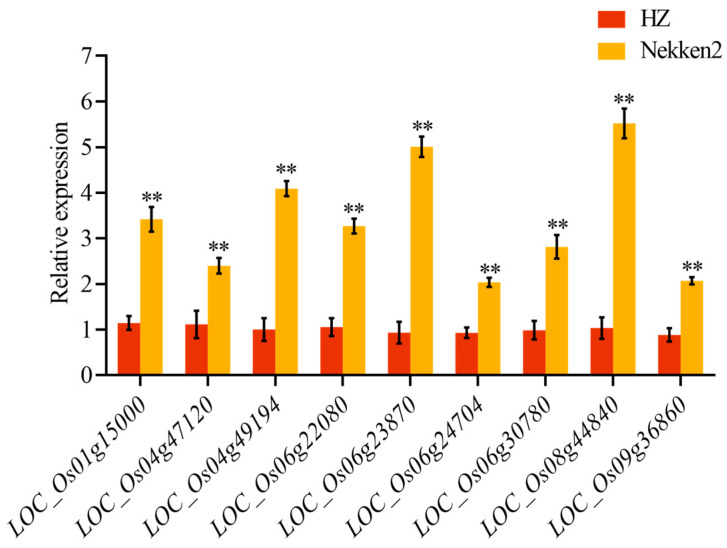
Analysis of differences in expression levels of fatty acid-related candidate genes in rice. ** in the *t*-text results, *p* < 0.01.

**Table 1 genes-15-01372-t001:** Sequence of qRT-PCR forward and reverse primer for candidate genes for fatty acid composition QTLs.

Primer Name	Sequence	Melting Temperature (°C)	Length (bp)
*LOC_Os01g15000*-F-qrt	CCATGTCATGTCGTCTGCTG	59.00	203
*LOC_Os01g15000*-R-qrt	TTTCTTCCTTGGCAGCAACC	58.96	
*LOC_Os04g47120*-F-qrt	GGTCGATATCTTCCGTGGGT	58.96	223
*LOC_Os04g47120*-R-qrt	TCTGGTGGCAAAGCTTGTTC	58.97	
*LOC_Os04g49194*-F-qrt	TGGCTGTGAACTACTACCCG	59.11	201
*LOC_Os04g49194*-R-qrt	CCTGTATCTGGTCGCCTAGG	59.04	
*LOC_Os06g22080*-F-qrt	AGTCTGGCTGCCCTTTAGTT	58.93	161
*LOC_Os06g22080*-R-qrt	CGTTGGGAAAGGGATTGGTG	59.11	
*LOC_Os06g23870*-F-qrt	GGGAAGGCATGGATCACAAA	58.15	309
*LOC_Os06g23870*-R-qrt	CGTTTACTTCGTAGCTGCCC	59.00	
*LOC_Os06g24704*-F-qrt	AAAGCGCCACCATGACAAAT	59.03	198
*LOC_Os06g24704*-R-qrt	TTAGCAGCTCCACCAATCCA	59.01	
*LOC_Os06g30780*-F-qrt	ATGGTGATGGAGGAGGCAC	57.89	173
*LOC_Os06g30780*-R-qrt	CGGCCGGAGAGATACATGTA	55.00	
*LOC_Os08g44840*-F-qrt	CCTTCTCCTCCTTCCAGTCG	59.18	160
*LOC_Os08g44840*-R-qrt	TGGATGAGGTTGCCGAAGTA	58.73	
*LOC_Os09g36860*-F-qrt	GCAGCCAGTGACACAAAGAT	58.76	157
*LOC_Os09g36860*-R-qrt	ATCCAGGGAATCAGCACCAA	59.00	

**Table 2 genes-15-01372-t002:** Fatty acid content correlation QTL analysis.

Content	QTL	Chromosome	Physical Distance (bp)	Position of Support (cM)	LOD
C_14_	*qCOFF4*	4	28,676,688~30,847,136	122.93~132.23	4.05
	*qCOFF6*	6	9,489,645~18,153,883	40.67~77.82	3.59
C_16:0_	*qCOFS4*	4	28,944,387~30,378,572	124.06~130.22	4.56
	*qCOFS6.1*	6	5,879,624~6,161,115	25.14~26.41	3.07
	*qCOFS6.2*	6	8,825,862~21,147,327	37.83~90.65	4.40
C_18:1_	*qCOPO1*	1	8,198,434~8,921,562	35.14~38.24	3.15
	*qCOPO3*	3	29,806,859~30,262,983	127.77~129.72	3.05
	*qCOPO4*	4	28,601,660~30,847,136	122.60~132.23	5.04
	*qCOPO6*	6	8,938,061~18,179,181	38.31~77.92	3.83
	*qCOPO8*	8	27,954,111~28,268,219	119.83~121.17	3.08
	*qCOPO9*	9	21,255,492~21,490,708	91.11~92.12	2.86
C_18:2_	*qCOPT4.1*	4	27,638,650~28,380,156	118.48~121.65	2.60
	*qCOPT4.2*	4	28,747,360~30,847,136	123.23~132.23	5.22
	*qCOPT6*	6	9,489,645~10,074,318	40.67~43.64	2.76

**Table 3 genes-15-01372-t003:** Fatty acid content determining candidate genes.

Chromosome	Gene ID	Function	Cloned or Not
1	*LOC_Os01g15000*	Lipase	Not cloned
4	*LOC_Os04g47120*	Acyl-CoA thioesterase 2	Not cloned
4	*LOC_Os04g49194*	Flavanone 3-hydroxylase	Not cloned
6	*LOC_Os06g22080*	Diacylglycerol O-acyltransferase	Not cloned
6	*LOC_Os06g23870*	Acyl-CoA dehydrogenase domain protein	Not cloned
6	*LOC_Os06g24704*	Rice acyl-CoA oxidase gene	Cloned
6	*LOC_Os06g30780*	Acyl-desaturase, chloroplast precursor	Not cloned
8	*LOC_Os08g44840*	BAHD acyltransferase-like protein gene; slender grain dominant	Cloned
9	*LOC_Os09g36860*	Acyl carrier protein	Not cloned

## Data Availability

The original contributions presented in the study are included in the article material, and further inquiries can be directed to the corresponding authors.

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
