# Peer review of "Genetic Loci Mining and Candidate Gene Analysis for Determining Fatty Acid Composition in Rice"

_genes, 2024, doi:10.3390/genes15111372_

Round 1
Reviewer 1 Report
Comments and Suggestions for Authors
The manuscript presents a study on genetic loci mining and candidate gene analysis related to fatty acid content in rice. The research is timely and relevant, given the increasing focus on improving the nutritional value of staple crops like rice. The integration of QTL mapping and candidate gene expression analysis is commendable, and the results may have potential applications in rice breeding programs aiming to enhance the fatty acid profile of rice for better nutrition and storage properties.
However, there are some aspects of the manuscript that need to be edited and supplemented to increase the accuracy of the published data.
1. Check the order of citations for correctness as it appears that citation [1] is missing from the manuscript.
2. In the Fatty Acid Content Determination description, it does not show whether you have technique replication measurements or not. Raw data of Fatty Acid Content should be adding in supplemental data to increase measurement validity and reliability.
3. Data from single measurements, one crop, or one location are insufficient for robust QTL mapping in the case of traits like fatty acid content, which are highly influenced by environmental conditions.
4. In figure 2: Whether the identified QTLs overlap with previously reported QTLs or whether they represent entirely novel findings? Including previously reported QTLs in the comparison will strengthen your argument about the novelty and relevance of your findings.
5. There is a big gap of the transition from detected QTLs to candidate genes. Why were these specific genes chosen for further analysis?
6. Differential gene expression analysis between two parental lines is good but not enough. The two parental lines (indica and japonica) belong to genetically distinct subspecies of rice. The gene expression differences observed in the parental lines may be influenced by their broad genetic background, not just the QTLs linked to fatty acid content. This makes it difficult to attribute the differential expression solely to the target QTL regions controlling fatty acid traits. Additional RILs that show significant differences in fatty acid content should be included. These selected RILs should remain genetically similar at non-target regions, which would allow for more precise identification of genes underlying the fatty acid trait.
Author Response
Dear Expert:
Thank you for your comments and suggestions on our submitted manuscript (genes-3241464); these suggestions helped us to substantially improve the manuscript. We have resubmitted the manuscript after extensive revisions and appreciate your willingness to review it again, ‘Genetic loci mining and candidate gene analysis determining fatty acid composition in rice’ to be published in genes.
In the revised manuscript, we have responded, provided additional explanations, and analysed the concerns of you, including more details and adding more validation. Here, we briefly summarise them as follows:
Comments 1: Check the order of citations for correctness as it appears that citation [1] is missing from the manuscript.
Response 1: Thank you for your review, the missing citations were an oversight when we were combing through the content of the article, we have re-combined and checked the content and order of the references in this manuscript and revised them in the manuscript.
Comments 2: In the Fatty Acid Content Determination description, it does not show whether you have technique replication measurements or not. Raw data of Fatty Acid Content should be adding in supplemental data to increase measurement validity and reliability.
Response 2: Thank you for this valuable suggestion. We divided our measurements into two parallel groups, each containing data from 8 parental lines and 120 recombinant inbred lines (RILs). Six fatty acids (C14:0, C16:0, C17:0, C18:0, C18:1, and C18:2) were measured. The population traits were stable, and the data reliable. Due to the large volume of raw data, we have uploaded it as an attachment rather than including it as supplementary data in the manuscript.
Comments 3: Data from single measurements, one crop, or one location are insufficient for robust QTL mapping in the case of traits like fatty acid content, which are highly influenced by environmental conditions.
Response 3: Thank you for raising this important point, it is a good question and I will respond to you with an explanation. After years of investigation and characterisation of the genotypes and phenotypes of this population at different locations (Hangzhou and Hainan) and in different seasons, it was found that the phenotypes were almost unchanged under the influence of the environment. The construction, genotypic and phenotypic investigations of this population were carried out in the same way as in the article ‘A strigolactone biosynthesis gene contributed to the green revolution in rice’ on Mol Plant, where we used the same methodology and a population of 120 RILs, which provides detailed information about the population, which we also cite in this paper.
Comments 4: In figure 2: Whether the identified QTLs overlap with previously reported QTLs or whether they represent entirely novel findings? Including previously reported QTLs in the comparison will strengthen your argument about the novelty and relevance of your findings.
Response 4: Thank you for your suggestion that the QTL embodied in Figure 2 of this manuscript overlap with previously reported QTLs with high LOD values and good confidence. These contents have been reflected in the discussion section of the manuscript, and in this revision, we have added more relevant contents in the discussion and made analyses in order to strengthen the relevance and credibility of the manuscript contents.
Comments 5: There is a big gap of the transition from detected QTLs to candidate genes. Why were these specific genes chosen for further analysis?
Response 5: Thanks for your question, in QTL research, screening candidate genes is an important step to understand the genetic basis of traits. In this study, we first determined the location of fatty acid content related QTL localisation interval results on the chromosome, combined with the rice genome annotation website (http://rice.plantbiology. msu.edu/) to find, screen and determine the We firstly identified the location of the QTL intervals on the chromosome, and combined with the rice genome annotation website, we searched, screened and identified the functional genes in the open reading frames corresponding to the QTL intervals, and consulted the literature for a preliminary summary of the candidate genes that might be related to fatty acid content of rice, and then combined with qRT-PCR experiments to screen out the genes that had large differences in the expression amounts in HuaZhan and Nekken2.
Comments 6: Differential gene expression analysis between two parental lines is good but not enough. The two parental lines (indica and japonica) belong to genetically distinct subspecies of rice. The gene expression differences observed in the parental lines may be influenced by their broad genetic background, not just the QTLs linked to fatty acid content. This makes it difficult to attribute the differential expression solely to the target QTL regions controlling fatty acid traits. Additional RILs that show significant differences in fatty acid content should be included. These selected RILs should remain genetically similar at non-target regions, which would allow for more precise identification of genes underlying the fatty acid trait.
Response 6: Thank you for your comments and I strongly agree with the points you mentioned. Indeed, the observed differences in gene expression between the two parental strains (indica and japonica) could be influenced by a wide range of genetic backgrounds. In future studies, I will consider selecting RILs exhibiting significant differences with similar genetic backgrounds to reduce the interference of background effects. Meanwhile, we also plan to more accurately correlate gene expression with fatty acid content through multiple QTL analyses and functional validation. we look forward to further discussing with you on how we can improve our study design and analysis methods to enhance the quality of our research.
Finally, we appreciate you again for all your helpful comments and hope that the new version of the manuscript is suitable for publication. We look forward to hearing from you.
Yours faithfully,
Yiyun Ge, Xuan Li, Yiting Wei, Yuchun Rao;
College of Life Sciences, Zhejiang Normal University, Jinhua 321004, China;
Email: ryc@zjnu.cn.
Reviewer 2 Report
Comments and Suggestions for Authors
Dear Authors,
Reviewer comments genes-3241464
The manuscript entitled „Genetic loci mining and candidate gene analysis of fatty acid composition in rice“ represents a useful study aimed at an identification of quantitative trait loci (QTLs) and candidate genes, respectively, determining the content of four kinds of fatty acids in rice grains.
However, I have some comments on the present version of the manuscript which are provided below:
1/ In the manuscript title, I would recommend to replace the word „of“ with „determining“ as follows: „genetic loci mining and candidate gene analysis determining fatty acid composition in riceů.
2/ In Materials and methods, part 2.1. Plant materials, the sources of all plant materials used in the study, namely both parental genotypes Huazhan and Nekken2 have to be given, i.e., it has to be specified from which institution they were obtained.
3/ In Materials and methods, part 2.2.4. and further, the supplier of all chemicals used in the study has to be given.
4/ In Table 1, the word „information“ has to be removed from the legend, i.e., Table 1 legend has to be modified as follows: „Sequence of qRT-PCR forward and reverse primers for candidate genes for fatty acid composition QTL.“ Moroever, remove „(5´-3´)“ and leave just „Sequence“ in Table 1 heading since both forward and reverse primer sequences are listed in Table 1.
5/ Table 3: Modify the Table 3 legend as follows: „Fatty acid content determining candidate genes“ since there is a list of candidate genes, not QTLs, provided in Table 3. Moreover, in Table 3, modify the statement „Not cloning“ to „Not cloned“.
6/ In Figure 3 legend, the kind of statsitical test used for the determination of significant differences has to be given! Most probably, Student T-test was used for the determination of significant differences; however, it has to be written in the figure legend and also in Materials and methods section!
7/ Formal comments on the text related to English language and style:
In the whole manuscript, add a space between the word and the following reference!
Introduction, line 36: Correct the typing error in the word „the population“ (not „the populace“).
Results, line 196: Add the word „finding“ following „This“ in the statement: „This finding highlights that the synthesis and metabolism of rice fatty acids are regulated…“
Results, line 219: Remove „the“ preceding the word „rice“ in the statement: „…thereby affecting the nutritional quality of rice.“
Comments on the Quality of English Language
Dear Authors,
Reviewer comments genes-3241464
The manuscript entitled „Genetic loci mining and candidate gene analysis of fatty acid composition in rice“ represents a useful study aimed at an identification of quantitative trait loci (QTLs) and candidate genes, respectively, determining the content of four kinds of fatty acids in rice grains.
However, I have some comments on the present version of the manuscript which are provided below:
1/ In the manuscript title, I would recommend to replace the word „of“ with „determining“ as follows: „genetic loci mining and candidate gene analysis determining fatty acid composition in riceů.
2/ In Materials and methods, part 2.1. Plant materials, the sources of all plant materials used in the study, namely both parental genotypes Huazhan and Nekken2 have to be given, i.e., it has to be specified from which institution they were obtained.
3/ In Materials and methods, part 2.2.4. and further, the supplier of all chemicals used in the study has to be given.
4/ In Table 1, the word „information“ has to be removed from the legend, i.e., Table 1 legend has to be modified as follows: „Sequence of qRT-PCR forward and reverse primers for candidate genes for fatty acid composition QTL.“ Moroever, remove „(5´-3´)“ and leave just „Sequence“ in Table 1 heading since both forward and reverse primer sequences are listed in Table 1.
5/ Table 3: Modify the Table 3 legend as follows: „Fatty acid content determining candidate genes“ since there is a list of candidate genes, not QTLs, provided in Table 3. Moreover, in Table 3, modify the statement „Not cloning“ to „Not cloned“.
6/ In Figure 3 legend, the kind of statsitical test used for the determination of significant differences has to be given! Most probably, Student T-test was used for the determination of significant differences; however, it has to be written in the figure legend and also in Materials and methods section!
7/ Formal comments on the text related to English language and style:
In the whole manuscript, add a space between the word and the following reference!
Introduction, line 36: Correct the typing error in the word „the population“ (not „the populace“).
Results, line 196: Add the word „finding“ following „This“ in the statement: „This finding highlights that the synthesis and metabolism of rice fatty acids are regulated…“
Results, line 219: Remove „the“ preceding the word „rice“ in the statement: „…thereby affecting the nutritional quality of rice.“
Author Response
Dear Expert:
Thank you for your comments and suggestions on our submitted manuscript (genes-3241464); these suggestions helped us to substantially improve the manuscript. We have resubmitted the manuscript after extensive revisions and appreciate your willingness to review it again, ‘Genetic loci mining and candidate gene analysis determining fatty acid composition in rice’ to be published in genes.
In the revised manuscript, we have responded, provided additional explanations, and analysed the concerns of you, including more details and adding more validation. Here, we briefly summarise them as follows:
Comments 1: In the manuscript title, I would recommend to replace the word “of”with “determining”as follows: “genetic loci mining and candidate gene analysis determining fatty acid composition in rice.”
Response 1: Thank you for your suggestions, which are good, and we have updated the title accordingly.
Comments 2: In Materials and methods, part 2.1 Plant materials, the sources of all plant materials used in the study, namely both parental genotypes Huazhan and Nekken2 have to be given, i.e., it has to be specified from which institution they were obtained.
Response 2: Thank you for your suggestion. HuaZhan, as an indica-type conventional rice, was selected and bred by the China Rice Research Institute and the Rice Research Institute of the Guangdong Academy of Agricultural Sciences, and has good growth characteristics and adaptability; whereas, Nekken2, as a japonica-type conventional rice, was selected and bred by Japan, and has broad-spectrum affinity and strong affinity. The parental varieties ‘HuaZhan’ and ‘Nekken2’ and the recombinant inbred population materials used in this study were provided by the China Rice Research Institute, established jointly with the group and grown for a long period of time.
Comments 3: In Materials and methods, part 2.2.4. and further, the supplier of all chemicals used in the study has to be given.
Response 3: Thank you for your suggestion, we have added the suppliers of the chemicals used in the study to the manuscript.
Comments 4: In Table 1, the word “information” has to be removed from the legend, i.e., Table 1 legend has to be modified as follows: “Sequence of qRT-PCR forward and reverse primers for candidate genes for fatty acid composition QTL.” Moroever, remove “(5´-3´)” and leave just“Sequence”in Table 1 heading since both forward and reverse primer sequences are listed in Table 1.
Response 4: Thank you for your suggestion, we have made the requested changes in the manuscript.
Comments 5: Table 3: Modify the Table 3 legend as follows: “Fatty acid content determining candidate genes”since there is a list of candidate genes, not QTLs, provided in Table 3. Moreover, in Table 3, modify the statement “Not cloning” to “Not cloned”.
Response 5: Thank you for your suggestion, we have made changes to the table in the manuscript.
Comments 6: In Figure 3 legend, the kind of statsitical test used for the determination of significant differences has to be given! Most probably, Student T-test was used for the determination of significant differences; however, it has to be written in the figure legend and also in Materials and methods section!
Response 6: Thank you for your question, we have spelled out the type of statistical test and its representation both in the experimental methods section and in the legend of Figure 3.
Comments 7: Formal comments on the text related to English language and style:
In the whole manuscript, add a space between the word and the following reference!
Introduction, line 36: Correct the typing error in the word “the population” (not “the populace”).
Results, line 196: Add the word “finding” following “This”in the statement: “This finding highlights that the synthesis and metabolism of rice fatty acids are regulated…”
Results, line 219: Remove “the”preceding the word “rice”in the statement: “…thereby affecting the nutritional quality of rice.”
Response 7: Thank you for your review and suggestions regarding the English language and style of the text, we have made changes in the manuscript and rechecked the spelling and expression of words and phrases in other parts of the text in accordance with the issues you raised.
Finally, we appreciate you again for all your helpful comments and hope that the new version of the manuscript is suitable for publication. We look forward to hearing from you.
Yours faithfully,
Yiyun Ge, Xuan Li, Yiting Wei, Yuchun Rao;
College of Life Sciences, Zhejiang Normal University, Jinhua 321004, China;
Email: ryc@zjnu.cn.
Round 2
Reviewer 1 Report
Comments and Suggestions for Authors
This is the follow up the previous comments and response:
Comments 1: The missing citation was added.
Comments 2: The data from supplemental file was not consistence with the data from the manuscript. The parental line name "RY" did not appear in the manuscript and the data of HZ and RY were not corresponding for HZ and Nekken2. Additionally, seven parental lines and four fatty acids were measured according to the table.
Comments 3: Based on the raw data from attachment, there is a big variation of between the lines of HZ even though they are purebred lines and cultivated at the same time and same location. Therefore, the response 3 mentioned that "the phenotypes were almost unchanged under the influence of the environment" is unfounded and contradicts what the author has discussed in the manuscript (Lines 246-249).
Comments 4: These contents were improved
Comments 5: It could be a relevant way to narrow down the candidate genes in this case.
Comments 6: Because previous steps such as QTL mapping and gene selection were based on insufficiently robust data, more reliable additional data are needed to substantiate the previous findings. The data of gene expression with the addition of the RILs are required.
Author Response
Dear Expert:
Thank you for your careful and comprehensive review of this manuscript (gene-3241464). It is your patient questions and revisions that make this study more scientific and complete. Below are our responses to your comments and corresponding revisions. We look forward to your approval:
Comments 2: The data from supplemental file was not consistence with the data from the manuscript. The parental line name "RY" did not appear in the manuscript and the data of HZ and RY were not corresponding for HZ and Nekken2. Additionally, seven parental lines and four fatty acids were measured according to the table.
Response 2: Thank you for your careful review. This is our oversight. "RY" in the original data is the Chinese abbreviation for the Nekken2 variety. We have updated the data table and the name in it to Nekken2. Our original data includes the four fatty acid contents of 136 strains (Sheet 1 ) including parental plants. Among them, to exclude the influence of growth environment and various biological stresses on a single parent, we planted a pair of parental plants every few RILs, including the beginning, for a total of eight pairs. Sheet 2 reflects the data from the eight pairs of parents and 120 RILs, listed separately, divided into two parallel groups, and then the average value is taken to determine the QTL interval. The data is comprehensive and the results are true and reliable.
Comments 3: Based on the raw data from attachment, there is a big variation of between the lines of HZ even though they are purebred lines and cultivated at the same time and same location. Therefore, the response 3 mentioned that "the phenotypes were almost unchanged under the influence of the environment" is unfounded and contradicts what the author has discussed in the manuscript (Lines 246-249).
Response 3: Thank you for your questions about the manuscript. To eliminate the differences you mention, we planted eight groups of parental plants, analysed the data, took the average and then determined the QTL. Regarding the contradiction between the previous round of replies and the discussion part of the manuscript, what we wanted to express to you in the previous round of replies was that our group had been planted in two places for many years and the traits were stable and reliable. In the discussion part, we wanted to express that the fatty acid content of plants varies greatly in different environments and conditions. We do not think this is a contradiction. In this revision we have also changed the wording of the manuscript to avoid any misunderstanding. Thanks again for your thoughtful consideration.
Comments 6: Because previous steps such as QTL mapping and gene selection were based on insufficiently robust data, more reliable additional data are needed to substantiate the previous findings. The data of gene expression with the addition of the RILs are required.
Response 6: Thank you for your question. As you mentioned, the QTL mapping itself is a relatively rough positioning, which is intended to provide clues for the subsequent study of the control gene for fatty acid content. We will build a backcross population based on the results of the QTL mapping and then carry out further cloning. We have combined some of the annotation functions on the rice data website, and this study also tried to do expression analysis of these possible candidate genes to determine the target gene more quickly. Thanks again for your suggestions.
Finally, we would like to thank you again for your comments on this manuscript. Your comments and suggestions took this study to an advanced level and gave us guidance for our future research. It is our hope that the new version of the manuscript will be suitable for publication. We are looking forward to receiving your response.
Yours faithfully,
Yiyun Ge, Xuan Li, Yiting Wei, Yuchun Rao;
College of Life Sciences, Zhejiang Normal University, Jinhua 321004, China;
Email: ryc@zjnu.cn.
2024.10.15